# Rapid and Sensitive Detection of *Verticillium dahliae* from Soil Using LAMP-CRISPR/Cas12a Technology

**DOI:** 10.3390/ijms25105185

**Published:** 2024-05-10

**Authors:** Yuxiao Fang, Lijuan Liu, Wenyuan Zhao, Linpeng Dong, Lijuan He, Yuhan Liu, Jinyao Yin, Yufang Zhang, Weiguo Miao, Daipeng Chen

**Affiliations:** Key Laboratory of Green Prevention and Control of Tropical Plant Diseases and Pests, Ministry of Education and School of Tropical Agriculture and Forestry, Hainan University, Haikou 570228, China; 21210904000003@hainanu.edu.cn (Y.F.); 17798439527@163.com (L.L.); 13233914129@163.com (W.Z.); donglinpeng@163.com (L.D.); helijuan741@163.com (L.H.); 21110710000024@hainanu.edu.cn (Y.L.); 20071000110015@hainanu.edu.cn (J.Y.); zhangyuf2023@163.com (Y.Z.); miao@hainanu.edu.cn (W.M.)

**Keywords:** Verticillium wilt, *V. dahliae*, soil-borne, LAMP-CRISPR/Cas12a

## Abstract

Cotton Verticillium wilt is mainly caused by the fungus *Verticillium dahliae*, which threatens the production of cotton. Its pathogen can survive in the soil for several years in the form of microsclerotia, making it a destructive soil-borne disease. The accurate, sensitive, and rapid detection of *V. dahliae* from complex soil samples is of great significance for the early warning and management of cotton Verticillium wilt. In this study, we combined the loop-mediated isothermal amplification (LAMP) with CRISPR/Cas12a technology to develop an accurate, sensitive, and rapid detection method for *V. dahliae*. Initially, LAMP primers and CRISPR RNA (crRNA) were designed based on a specific DNA sequence of *V. dahliae*, which was validated using several closely related *Verticillium* spp. The lower detection limit of the LAMP-CRISPR/Cas12a combined with the fluorescent visualization detection system is approximately ~10 fg/μL genomic DNA per reaction. When combined with crude DNA-extraction methods, it is possible to detect as few as two microsclerotia per gram of soil, with the total detection process taking less than 90 min. Furthermore, to improve the method’s user and field friendliness, the field detection results were visualized using lateral flow strips (LFS). The LAMP-CRISPR/Cas12a-LFS system has a lower detection limit of ~1 fg/μL genomic DNA of the *V. dahliae*, and when combined with the field crude DNA-extraction method, it can detect as few as six microsclerotia per gram of soil, with the total detection process taking less than 2 h. In summary, this study expands the application of LAMP-CRISPR/Cas12a nucleic acid detection in *V. dahliae* and will contribute to the development of field-deployable diagnostic productions.

## 1. Introduction

Cotton is an important economic crop worldwide, contributing approximately 35% of the world’s total natural fiber to the textile industry. Additionally, it serves as a source of edible oil and livestock feed [1]. Verticillium wilt is a significant vascular soil-borne disease affecting cotton. On average, it results in yield losses ranging from approximately 10% to 35% [2]. In China, cotton production has suffered substantial economic losses due to the widespread occurrence of Verticillium wilt, affecting approximately 2.5 million hectares of cotton annually [3]. The primary causal agent of cotton Verticillium wilt is the soil-borne pathogenic fungus *Verticillium dahliae* [4]. The fungus *V. dahliae* produces a dormant structure called microsclerotia, which can survive in the soil for several years [5]. Microsclerotia serve as primary infection sources of cotton Verticillium wilt; when encountering a suitable host and environment, they germinate and produce infection hyphae to infect the roots of cotton plants. Once the fungus successfully invades cotton roots, its mycelium reaches the plant’s vascular tissue [5]. Subsequently, the mycelium produces a large number of spores, facilitating rapid vertical transmission within the vascular bundles. Simultaneously, the mycelium achieves swift horizontal transmission through the pits between the xylem vessels [6]. Therefore, rapid, sensitive, and accurate detection of *V. dahliae* in the soil is paramount for early monitoring, warning, and management of cotton Verticillium wilt [7,8].

The rapid diagnosis of cotton Verticillium wilt is crucial for its prediction and control. Various approaches have been reported for *V. dahliae* detection, including Droplet Digital PCR (ddPCR) [9], Quantitative Real-time PCR (qRT-PCR) [9,10,11], and Loop-Mediated Isothermal Amplification (LAMP) [12]. However, these methods also have their disadvantages. For example, ddPCR is costly and requires professionally trained technicians. Additionally, LAMP exhibits a high false-positive rate. Currently, there is a lack of a low-cost, easy-to-operate field detection methods for detection of the *V. dahliae* in soil.

The development of isothermal amplification techniques, such as Loop-Mediated Isothermal Amplification (LAMP), recombinase polymerase amplification (RPA), and rolling circle amplification (RCA), have successfully eliminated the requirement for thermocycling instruments and can be applied for real-time detection, thereby facilitating field and point-of-care testing. However, these methods may result in false-positive detection due to nonspecific amplification, cross-contamination, or primer dimerization [13,14]. As results, there remains a necessity to create innovative diagnostic platforms that enable the rapid, highly sensitive, and extremely specific detection of nucleic acids.

Recently, nucleic acid detection methods utilizing the clustered regularly interspaced short palindromic repeats (CRISPR)-associated endonucleases (CRISPR/Cas) systems have been developed [15,16,17,18,19]. In the type V-A CRISPR system, CRISPR/Cas12a (Cpf1) exhibits trans-cleavage activity in addition to its specific cleavage of double-stranded DNA (dsDNA) using a single guide RNA (sgRNA), resulting in cleavage of single-stranded probes for nucleic acid detection [17]. This approach has been implemented in SHERLOCK (Cas13) [16,20,21], DETECTR (Cas12 or Cas14) [15,22], and HOLMES (Cas12) [23]. The discovery of trans-cleavage activity has paved the way for a new generation of rapid nucleic acid detection techniques. By combining this with the isothermal amplification RPA assay, the CRISPR system has been utilized for *V. dahliae* detection [24]. However, the high cost of the RPA assay hinders its widespread application in field detection. Currently, there is no method available for integrating LAMP isothermal amplification technology with the CRISPR system for detecting *V. dahliae*. Proper primer design and selection play a crucial role in optimizing LAMP reactions [25]. Although LAMP amplification technology offers the advantage of high sensitivity, it also presents the drawback of a high false-positive rate. However, combining LAMP with the CRISPR system (LAMP-CRISPR/cas12a) can effectively mitigate this limitation. The combination of LAMP and CRISPR/Cas12a technology has been widely applied in the detection of plant pathogens [26,27,28,29,30], indicating that this technology has become quite mature.

A paper-based lateral flow strips (LFS) method, which combines the colloidal gold-based nanoparticles with conventional chromatographic separation, provides a new direction for rapid point-of-care (POC) diagnostics. These methods offer the advantages of being user-friendly, fast, and requiring no complicated instrumentation [31,32,33,34].

In this study, based on LAMP-CRISPR/cas12a technology, we developed a fluorescence visualization detection system, suitable for laboratory use, that requires only ultraviolet flashlight irradiation to enable the observation of results. Additionally, this study innovated a visual on-site detection system for *V. dahliae* by integrating field nucleic acid extraction, LAMP-CRISPR/cas12a and a portable lateral flow strip (LFS). Both devised systems attributes of economic feasibility, operational simplicity, heightened sensitivity, and specificity, along with visualization properties [35,36], all of which significantly to real-time monitoring and prompt establishment of control strategies against *V. dahliae*.

## 2. Results

### 2.1. Candidate DNA Fragment-Specific Analysis

To find a candidate target genomic DNA sequence of *V. dahliae* for LAMP-CRISPR/Cas12a detection, a 916 bp DNA sequence (CP010981.1:12,621–13,536) in *V. dahliae* was aligned with the whole genome sequences of 43 strains of *V. dahliae* and its closely related species in the NCBI database using BIO EDIT [37]. The results revealed the presence of a 178 bp sequence (CP010981.1:12,621–12,798) specifically in *V. dahliae* (Figure 1a). Furthermore, this sequence was validated in all 43 strains of *V. dahliae* by PCR using PCR-F/R primer pair (Appendix A) amplification from 16 representative strains of *V. dahliae* (Figure 1b). Based on this DNA segment, LAMP primers and CRISPR RNA (crRNA) were designed.

### 2.2. Specificity Test of LAMP-CRISPR/Cas12a Fluorescence Visualization Detection System

The specificity of the established LAMP-CRISPR/Cas12a assay was evaluated using representative *Verticillium* spp., bacterial and nine other fungi strains (*Verticillium longisporum*, *Verticillium alfalfa*, *Verticillium nubilum*, *Verticillium nigrescens*, *Verticillium alboatrum*, and *Verticillium nonalfalfae*; and *Xanthmonas citri* subsp. *malvacearum*, *Rhizoctonia solani*, *Trichoderma harzianum*, *Fusarium graminearum*, *Phytophthora capsici*, *Phytophthora sojae*, *Trichoderma asperellum*, *Fusarium oxysporum* f. sp. *Cubense*, and *Magnaporthe oryzae*). The results showed that the fluorescent signal could be detected from reactions of *V. dahliae*, but not from other bacterial and fungi strains (Figure 2a,b), indicating the specificity of the established LAMP-CRISPR/Cas12a system for *V. dahliae* detection. Subsequently, we validated the specificity of the established closed-tube detection method, and the results demonstrated its highly identical specificity to the open-tube detection method; the results of this method also showed that the fluorescent signal could be observed from reactions of *V. dahliae*, but not from other bacterial and fungi strains.

### 2.3. Sensitivity Test of LAMP-CRISPR/Cas12a Fluorescence Visualization Detection System

To test the sensitivity of LAMP-CRISPR/Cas12a, reaction was determined with a 10-fold serially diluted template at concentrations of 1 ng, 100 pg, 10 pg, 1 pg, 100 fg, 10 fg, 1 fg, and 100 ag of the genomic DNA template. The results showed that the developed LAMP–CRISPR/Cas12a system can detect less than 10 fg of the genomic DNA template (Figure 2c,d). Subsequently, sensitivity of the closed-tube detection method was validated; the results indicated that the optimized closed-tube detection method maintained the same sensitivity as the open-tube detection method.

### 2.4. Detection of V. dahliae in Complex Soil Samples Using LAMP-CRISPR/Cas12a Fluorescence Visualization Detection System

Since microsclerotia of *V. dahliae* are a primary source of fungal inoculum in field soil, it is imperative to evaluate the system for the detection of *V. dahliae* in soil using the established LAMP- CRISPR/Cas12a closed-tube fluorescence detection system. A serial dilution of soil containing one to 10 microsclerotia per 0.5 g of soil was prepared. The results showed that the LAMP-CRISPR/Cas12a fluorescence detection system was able to detect as few as 1 microsclerotium per 0.5 g of soil, indicating that the LAMP-CRISPR/Cas12a fluorescence detection system provided ultra-sensitive detection for complex soil samples. This fluorescence visualization detection system was also capable of detecting *V. dahliae* in natural disease soil in Xinjiang, China (Appendix A).

### 2.5. Development of a LAMP-CRISPR/Cas12a On-Site Detection System for V. dahliae in Soil

After extracting DNA using the aforementioned second soil DNA-extraction method (without requiring any additional equipment), we amplified the DNA using the established LAMP-CRISPR/Cas12a technology but replacing the ssDNA FQ probe with the ssDNA FB probe. The reaction was conducted using a smart thermos cup (HONGPA, Huawei, Shenzhen, China), which was connected to a smartphone through an app. The reaction procedure was as follows: LAMP amplification at 62 °C for 45 min followed by in vitro cleavage by CRISPRCas12a at 37 °C for 30 min. After the reaction, the portable lateral flow strip (LFS) (Warbio, Nanjing, China) was then inserted into the Cas12a reaction mixture for result interpretation (Figure 3). The specificity and sensitivity of the on-site LAMP-CRISPR/Cas12a system was also evaluated for the detection of *V. dahliae*. The results showed the fluorescent signal could only be detected for reactions with *V. dahliae* (Figure 4a,b). In addition, the developed LAMP–CRISPR/Cas12a on-site detection system can detect up to 1 fg of the genomic DNA (Figure 4c). Furthermore, the system possesses the sensitivity to detect as few as three microsclerotia per 0.5 g of soil. The entire process from DNA extraction to obtaining the test results takes less than 2 h. The established on-site detection system can also detect *V. dahliae* from natural disease soil in Xinjiang, China (Appendix A). 

## 3. Discussion

*V. dahliae* is a ubiquitous soil-borne fungal pathogen, capable of persisting in the soil in the form of microsclerotia for several years. Upon encountering suitable hosts or suitable environments, it can germinate and infect plants, causing Verticillium wilt and leading to significant yield decline in many economic crops. Although various nucleic acid detection techniques have been developed to detect *V. dahliae* in complex soil environments, the limitations of these methods hinder their widespread application. For instance, droplet digital PCR and LAMP technology exhibit high false-positive rates, while real-time fluorescence quantitative PCR is costly and complicated operation. Despite the high specificity and sensitivity of RPA-CRISPR/Cas12a, the high cost of RPA detection restricts field deployment. Currently, there remains a lack of a low-cost detection technology with strong specificity, high sensitivity, and suitability for on-site detection to detect *V. dahliae* in soil.

CRISPR/Cas12a-based nucleic acid detection technology is renowned for its exceptional specificity and sensitivity, rendering it suitable for pathogen detection in intricate soil samples. Although previous researchers have developed RPA-CRISPR/Cas12a detection techniques for *V. dahliae* in soil, the integration of LAMP with Cas12a remains unexplored. The LAMP technology necessitates four primers to identify six different regions on the target, representing a remarkably sensitive nucleic acid amplification method that employs bst polymerase, an enzyme highly resistant to PCR amplification inhibitors. By combining CRISPR/Cas12a, this approach transforms into an exceedingly specific and sensitive method for detecting soil-borne pathogens. Compared to RPA-CRISPR/Cas12a, the LAMP-CRISPR/Cas12a method exhibits reduced reliance on enzymes, requires fewer manual steps, and proves relatively more cost-effective, making it particularly suitable for on-site detection.

In this study, we developed a fluorescence detection system, specifically for *V. dahliae* in soil, based on a specific and intraspecific conserved sequence of *V. dahliae* combined with LAMP and CRISPR/Cas12a methods. For this system, we employed a method for the rapid extraction of *V. dahliae* DNA from soil using laboratory equipment. Additionally, by adding trehalose into the reaction mixture containing LAMP and Cas12a in the same tube, false-positive reactions were effectively avoided. Furthermore, to enhance the observation of nucleic acids in the CRISPR/Cas12a reaction, we screened probes with stronger fluorescence signals [27], making the system suitable for laboratory-based detection. The sensitivity of this system enables the detection of a single microsclerotium of *V. dahliae* in soil in less than 90 min from DNA extraction to detection.

Additionally, to facilitate on-site detection, we employed a self-established method for in situ extraction of *V. dahliae* DNA from soil; this involved integrating rapid DNA extraction and visualization with LAMP-CRISPR/Cas12a detection using an LFS device, thereby establishing an on-site detection system. The entire system can be operated in the field using a simple portable device, enabling the complete detection process to be completed within two hours. Although its sensitivity is slightly lower than that of the developed fluorometric detection system (which detects a minimum of three microsclerotia), it still fulfills the requirements for on-site testing. This study represents the first development of an on-site detection method for *V. dahliae* in soil.

Previously, pathogen DNA in soil was extracted either using expensive kits or through cumbersome and time-consuming steps requiring laboratory instruments, thus making on-site DNA extraction impossible. For the first time, this study established a low-cost and simple-operation LAMP-CRISPR/Cas12a detection method for rapid detection of *V. dahliae* in soil that meets all necessary requirements. Furthermore, this study provides valuable insights into fast detection methods for other soil-borne pathogens.

## 4. Materials and Methods

### 4.1. Materials and Nucleic Acid Extraction

The *V. dahliae* strains used in the experiment were provided by Xinjiang Academy of Agricultural Sciences and Jiangsu Academy of Agricultural Sciences, China (Appendix A). *V. longisporum* strains were provided by Gansu Academy of Agricultural Sciences. *V. alfalfae* and *V. nubilum* strains were provided by Northwest A&F University, China. *V. nigrescens*, *V. alboatrum*, *V. nonalfalfae* and *R. solani* strains were provided by Chinese Academy of Agricultural Sciences, China. The remaining plant pathogenic strains, including *T. harzianum*, *F. graminearum*, *P. capsici*, *Xanthmonas citri* subsp. *malvacearum* (*XCM*), *P. sojae*, *T. asperellum*, *Fusarium oxysporum f.* sp. *Cubense* (*FOC*) and *M. oryzae* are all preserved at Hainan University, China. The genomic DNA of the fungal strains was extracted using the Fungal DNA Kit (OMEGA BIOTEK, Norcross, GA, USA), while the bacterial strains underwent genomic DNA extraction via the TIANamp Bacteria DNA Kit (TIANGEN, Beijing, China).

All soil samples that did not contain *V. dahliae* and had no history of wilt disease were collected from the agricultural base of Hainan University in Hainan Province, China. The microsclerotia of *V. dahliae* was prepared according to the method reported by Pérez-Artés, et al. (2004) [38], and different quantities of microsclerotia were mixed into clean soil to prepare artificial infested soil. The natural disease soil used in this study was taken from around diseased cotton plants in Xinjiang, China, at a depth of approximately 20 cm.

Two methods of fungal DNA extraction from soil samples were used in this study. The first method requires the use of laboratory equipment; the details of the procedure were as follows: 0.5 g soil sample was added to a 2 mL centrifuge tube, which contained 600 μL lysis solution (100 mmol·L^−1^ Tris-HCl, 100 mmol·L^−1^ EDTA·Na_2_, 100 mmol·L^−1^ Na_3_PO_4_, 200 mmol·L^−1^ NaCl, 2% PVPP, 0.5% SDS, pH 8.0) and 2 approximately 5 mm steel beads. The tube was then shaken in a bead beater (60 hz, 100 s), centrifuged at 15,000 rpm/min for 5 s, and then filter paper strips (The Whatman No. 1 test paper (Whatman, Maidstone, UK) were used; before use, they were processed by first immersing half of the test paper into molten paraffin to form a hydrophobic zone. After the paraffin solidified, the partially paraffin-coated filter paper was cut into a 44 mm wide rectangle, with approximately 40 mm coated with paraffin and 4 mm uncoated. Then, this rectangle was cut into strips approximately 2 mm wide (forming test paper with a 2 × 4 mm nucleic acid binding region and a 2 × 40 mm handle) that were used to touch the supernatant three times, each time for 1–2 s, to bind the nucleic acids. The filter paper strips from the previous step were then dipped into 2 mL wash solution (10 mM Tris [pH 8.0], 0.1% Tween-20) three times, and after washing, the strips were dipped into PCR tubes containing 40 μL nuclease-free water three times, each time for 1–2 s. After mixing the 40 μL system, 1.5 μL was taken as the template for the LAMP-CRISPR/Cas12a reaction. The second method does not require the use of laboratory equipment and can allow for on-site soil DNA extraction: a 0.5 g soil sample was added to a 2 mL centrifuge tube, which contained 800 μL of the above lysis solution and steel beads. The tube was vigorously shaken for approximately 30 s for disruption, and then left to stand until 150–200 μL of supernatant appeared (this process takes about 10–30 min). The supernatant was then transferred to a new 1.5 mL centrifuge tube and left to stand for about 5 min for the precipitation of excess impurities, followed by using filter paper strips following the above method.

### 4.2. PCR Primers, LAMP Primers, and Reporter Probes

After conducting an analysis of a DNA sequence spanning 916 bp, it was revealed that a segment of 178 bp is unique to the *V. dahliae*. Comparing this segment to the DNA sequences of the same species and related species on NCBI, it was found that the 178 bp sequence varies significantly between different species of *Verticillium* but remains highly conserved within the species of *V. dahliae*. Firstly, based on the 916 bp DNA sequence, a PCR primer pair PCR-F/R with a target fragment of 600 bp was designed and used to amplify the genomic DNA of 16 representative strains of *V. dahliae* from Xinjiang and Jiangsu. The LAMP primers were designed based on the 178 bp sequence using Primer Explorer V5 and synthesized by Bgi Tech Solutions (Liu He) Co., Ltd. (Beijing, China). Five sets of primers were designed to amplify the genomic DNA of *V. dahliae*. One primer set was found to have good amplification efficiency within 45 min, the LAMP primer results are shown in Appendix A. For the fluorescent detection system, an ssDNA FQ probe (labeled with FAM and BHQ1) was designed. For the immune-assay strip detection reaction, an ssDNA FB probe (labeled with FAM and biotin) was designed; both probes were synthesized by Bgi Tech Solutions (Liu He) Co., Ltd.

### 4.3. Guide RNA Design and sgRNA Synthesis

The crRNA of Cas12a requires a specific TTTN PAM (protospacer adjacent motif), and, based on the selected LAMP sequences as targets, relevant crRNAs were designed using the Benchling online platform. Two high-scoring crRNAs sequences were chosen. Bgi Tech Solutions (Liu He) Co., Ltd. (Beijing, China) was commissioned to synthesize the corresponding Oligo short chains with T7 promoter, repeat sequences, and spacer sequences, which served as templates for the guide RNA. After synthesizing crRNA using the T7 Quick High Yield RNA Synthesis Kit (New England Biolabs, Cat#E2050S) and purifying it using the RNA Cleanup Kit (New England Biolabs, Cat#T2040S), the purified crRNA was diluted and stored at −80 °C for future use.

### 4.4. LAMP Reaction

The LAMP reaction was conducted by adding 2.5 μL of 10× ThermoPol Buffer, 1.5 μL of 100 mM MgSO_4_, 1.4 μL of 25 mM dNTPs, 40 μM of LAMP-FIP primer and LAMP-BIP primer, 5 μM of LAMP-F3 primer and LAMP-B3 primer, 10 μM of LAMP-LF primer and LAMP-LB primer each 1 μL, 1 μL of 8000 U/mL Bst 2.0 WarmStart DNA Polymerase (New England Biolabs, Ipswich, MA, USA), 1 μL of 5 mM betaine, and 1 μL of template DNA, with sterilized ultrapure water added to make up to 25 μL. The reaction was held at 62 °C for 45 min, and the LAMP products were analyzed by 2% agarose gel electrophoresis.

### 4.5. LAMP-CRISPR/Cas12a Fluorescence Visualization Detection System

First, the reaction system combining LAMP and CRISPR/Cas12a cleavage technology was established. The LAMP amplification technology was used to amplify the genome DNA of the bacterium in a PCR reaction tube. The LAMP system included 2.5 μL of 10× ThermoPol Buffer, 1.5 μL of 100 mM MgSO_4_, 1.4 μL of 25 mM dNTP, 4 μM of LAMP-FIP primer and LAMP-BIP primer, 5 μM of LAMP-F3 primer and LAMP-B3 primer, 10 μM of LAMP-LF primer and LAMP-LB primer each 1 μL, 1 μL of 8000 U/mL Bst 2.0 WarmStart DNA Polymerase,1 μL of 5 mM betaine, and 1 μL of template DNA, with sterilized ultrapure water added to make up to 25 μL. The Cas12a cleavage system was placed in another reaction tube, which included 3 μL 10× NEBuffer r2.1, 3 μL of 300 nM crRNA, 1 μL of 1 µM EnGen Lba Cas12a (Cpf1) (New England Biolabs), 0.5 μL of RNase inhibitor (TaKaRa, Dalian, China), 1 μL of ssDNA FQ probe group, and supplemented with enzyme-free water to a total volume of 27 μL. After the LAMP amplification was completed, 1 μL of LAMP product was taken and transferred to the Cas12a cleavage system for targeting DNA cleavage. Fluorescence intensity was measured using an enzyme maker (Infinite M200 PRO). The fluorescence signal of ssDNA-FQ report probe was visualized with a UV flashlight. Positive results (containing the genome DNA of the *V. dahliae*) can be observed as green fluorescence under UV flashlight, while negative results show no fluorescence.

However, the process of opening the lid of reaction tubes containing LAMP amplification during the reaction can lead to aerosol contamination, due to the large amount of LAMP products, which may result in false-positive outcomes. To address this issue, a strategy involving the addition of trehalose (Solarbio, Beijing, China) to the Cas12a system was employed to enhance the thermal instability of the Cas12a protein. The LAMP-CRISPR/Cas12a closed tube detection system is as follows: The LAMP system is placed at the bottom of the tube and the Cas12a cleavage system at the top. After the LAMP amplification is complete, the Cas12a system is centrifuged or manually shaken from the tube cap to mix the two systems. It is then incubated at 37 °C for 30 min in a constant-temperature water bath.

## Figures and Tables

**Figure 1 ijms-25-05185-f001:**
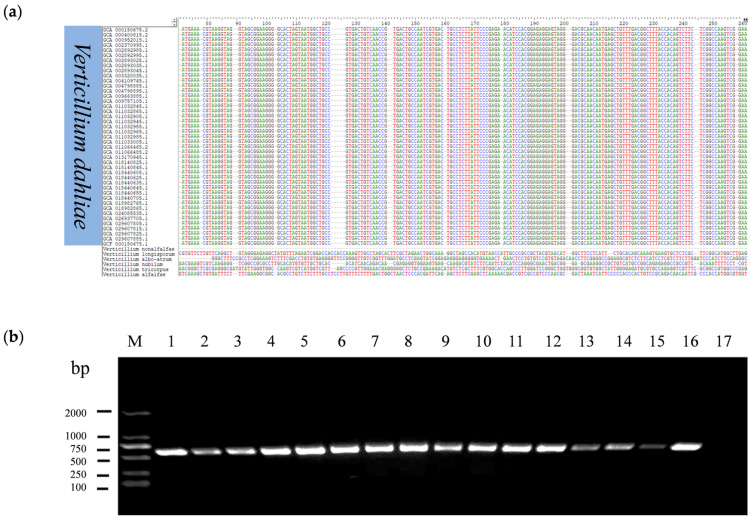
Candidate-specific analysis of DNA fragment from *V. dahliae.* (**a**): Multiple sequence alignment of candidate DNA sequences of *V. dahliae*. (**b**): PCR detection of 16 strains of *V. dahliae*. M: DL2000; 1–16: DNA of different strains of *V. dahlia*; 17: ddH_2_O.

**Figure 2 ijms-25-05185-f002:**
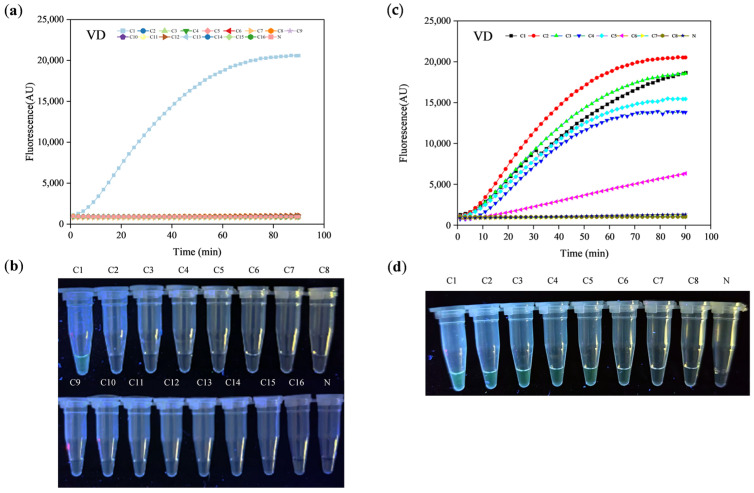
Establishment of the LAMP-CRISPR/Cas12a fluorescence visualization detection system. A and B: Specificity of the established detection system; (**a**): The fluorescent signals were calculated by enzyme maker; (**b**): Visualized under UV flashlight. C1: Positive (*V. dahliae* DNA); C2: *V. longisporum* DNA; C3: *V. alfalfae* DNA; C4: *V. nubilum* DNA; C5: *V. nigrescens* DNA; C6: *V. alboatrum* DNA; C7: *V. nonalfalfae* DNA; C8: *R. solani* DNA; C9: *T. harzianum* DNA; C10: *F. graminearum* DNA; C11: *P. capsici* DNA; C12: *XCM* DNA; C13: *P. sojae* DNA; C14: *T. asperellum* DNA; C15: *FOC* DNA: C16: *M. oryzae* DNA; N: Negative. (**c**,**d**): Sensitivity of the established detection system; (**a**): The fluorescent signals were calculated by enzyme maker; (**b**): Visualized under UV flashlight; C1: 1 ng; C2: 100 pg; C3: 10 pg; C4: 1 pg; C5: 100 fg; C6: 10 fg; C7: 1 fg; C8: 100 ag; N: Negative.

**Figure 3 ijms-25-05185-f003:**
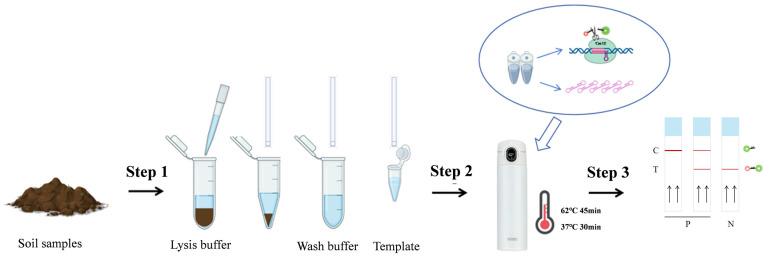
The operational workflow of the LAMP-CRISPR/Cas12a on-site detection system is delineated as follows: Step 1, an extraction of total DNA from soil is carried out; Step 2, the LAMP-CRISPR/Cas12a reaction is facilitated within a smart thermos cup; Step3, using a lateral flow strip (LFS) for result readout.

**Figure 4 ijms-25-05185-f004:**
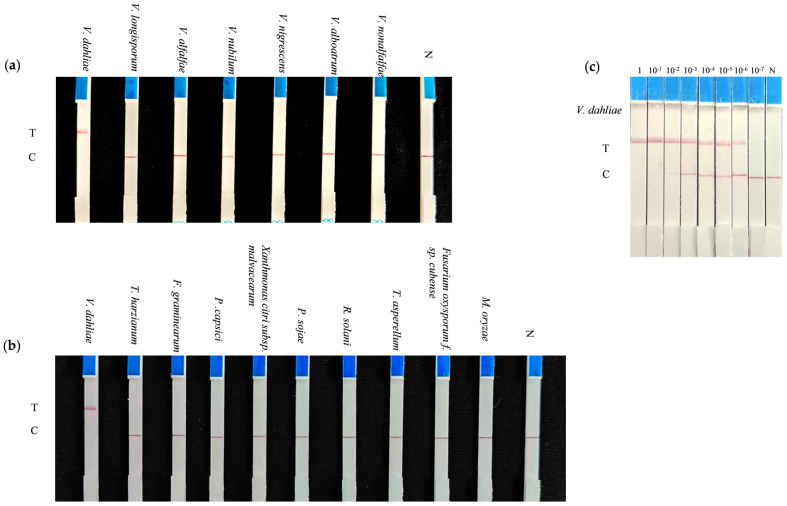
Establishment of the LAMP-CRISPR/Cas12a on-site detection system. (**a**,**b**): Specificity of this system, using a lateral flow strip (LFS) for result readout. (**c**): Sensitivity of this system, using a lateral flow strip (LFS) for result readout.

## Data Availability

The data that support the findings of this study are available from the corresponding author upon reasonable request.

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
