# Peer review of "Rapid and Sensitive Detection of Verticillium dahliae from Soil Using LAMP-CRISPR/Cas12a Technology"

_ijms, 2024, doi:10.3390/ijms25105185_

Round 1

Reviewer 1 Report

Comments and Suggestions for Authors

This ARTICLE entitled "Rapid and sensitive detection of Verticillium dahliae from soil using LAMP-CRISPR/Cas12a technology" by Fang et al develop the LAMP-CRISPR/Cas12a technology for V. dahliae detection. This study shows well-planned experimental results, and these results are thought to be of interest to readers. I believe the manuscript is suitable for publication in the IJMS after minor revisions.

Minor Concerns:

1.       The pathogenicity is different in different V. dahliae strains, can the author detect them?

2.       The discussion seems sketchy, the authors need to fully discuss the present results with previous studies.

3.       The reference format requires a lot of checking, many have no page numbers.

Comments on the Quality of English Language

The English is quite well.

Author Response

  1. The pathogenicity is different in different V. dahliae strains, can the author detect them?

RE: Thank you very much for taking the time to review this manuscript. Regarding your first question, as only morphological identification were performed on the V. dahliae  strains at the time, and no pathogenicity testing was conducted. 

  1. The discussion seems sketchy; the authors need to fully discuss the present results with previous studies.

RE: Thanks. As suggestion, I have made improve to the discussion section in the resubmitted manuscript.

  1. The reference format requires a lot of checking, many have no page numbers.

RE: Thanks. As suggestion the missing page numbers have been added to the reference.

Reviewer 2 Report

Comments and Suggestions for Authors

The current manuscript, entitled "Rapid and sensitive detection of Verticillium dahliae from soil using LAMP-CRISPR/Cas12a technology" focuses on rapid detection of soil pathogen V. dahliae.  The authors have shown the lower detection limit and time-saving approach of the developed LAMP-CRISPR/Cas12a-LFS system.

Overall, the study has novelty and significance. However, it needs to be improved in certain aspects. Here I am suggesting some of the changes that should be included in the manuscript.

1. Authors should include the full form of each of the abbreviations, used in the manuscript.

2. The full name of the fungal strains used in the study should be included in the first place.

3. The Discussion section is written very casually and, therefore, should be improved

through related and recent studies.

4. Authors are suggested to provide the company and country name of the materials used.

Author Response

  1. Authors should include the full form of each of the abbreviations, used in the manuscript.

RE: Thank you very much for taking the time to review this manuscript. Regarding your first question, I have made the necessary modifications.

  1. The full name of the fungal strains used in the study should be included in the first place.

RE: Thanks, revised as suggestion.

  1. The Discussion section is written very casually and, therefore, should be improvedthrough related and recent studies.

 RE: Thanks. As suggestion, I have made improve to the discussion section in the resubmitted manuscript.

  1. Authors are suggested to provide the company and country name of the materials used.

RE: Thanks. I have included the names of the companies and countries for the main reagents and materials used.

Reviewer 3 Report

Comments and Suggestions for Authors

After reading the manuscript extended for review, I believe that this is a very important work in terms of science and application, i.e. from the point of view of agriculture. Overall, the work is well organized and written. I found a few minor technical things to improve, which I've listed below. However, there is one weak point of this manuscript - the discussion. In my opinion, this section needs to be significantly improved.

My comments:

1) Ln 79: missing space “...g LAMP reactions[25].”

2) Ln 111: missing italics “...V. dahliae DNA..”

3) “2.2 Specificity test of LAMP-CRISPR/Cas12a fluorescence…” – missing italics in several Latin names

4) Discussion - I'm sorry, but it's hard to comment. I am asking the authors to improve this part of the manuscript.

5) Materials and Method – something wrong with the font size compared to the rest of the manuscript.

Author Response

  1. Ln 79: missing space “...g LAMP reactions[25].”

REThank you very much for taking the time to review this manuscript. Regarding your first question, I have made the necessary modifications.

  1. Ln 111: missing italics “...V. dahliae DNA..”

RE:  Thanks. Revised as suggestion.

  1. “2.2 Specificity test of LAMP-CRISPR/Cas12a fluorescence…” – missing italics in several Latin names.

RE: Thanks. Revised as suggestion.

  1. Discussion - I'm sorry, but it's hard to comment. I am asking the authors to improve this part of the manuscript.

RE: Thanks. As suggestion, I have made improve to the discussion section in the resubmitted manuscript.

  1. Materials and Method – something wrong with the font size compared to the rest of the manuscript.\

RE: Thanks. Revised as suggestion.